# Complement and Fungal Dysbiosis as Prognostic Markers and Potential Targets in PDAC Treatment

**Cornelia Speth** [1,*,†], **Ruben Bellotti** [2,†], **Georg Schäfer** [3], **Günter Rambach** [1], **Bernhard Texler** [2], **Gudrun C. Thurner** [3], **Dietmar Öfner** [2], **Cornelia Lass-Flörl** [1] and **Manuel Maglione** [2]

1   Institute of Hygiene and Medical Microbiology, Medical University of Innsbruck, 6020 Innsbruck, Austria
2   Department of Visceral, Transplant, and Thoracic Surgery, Medical University of Innsbruck, 6020 Innsbruck, Austria
3   Institute of Pathology, Neuropathology and Molecular Pathology, Medical University of Innsbruck, 6020 Innsbruck, Austria
*   Correspondence: cornelia.speth@i-med.ac.at; Tel.: +43-512-9003-70705
†   These authors contributed equally to this work.

**Abstract:** Pancreatic ductal adenocarcinoma (PDAC) is still hampered by a dismal prognosis. A better understanding of the tumor microenvironment within the pancreas and of the factors affecting its composition is of utmost importance for developing new diagnostic and treatment tools. In this context, the complement system plays a prominent role. Not only has it been shown to shape a T cell-mediated immune response, but it also directly affects proliferation and apoptosis of the tumor cells, influencing angiogenesis, metastatic spread and therapeutic resistance. This makes complement proteins appealing not only as early biomarkers of PDAC development, but also as therapeutic targets. Fungal dysbiosis is currently the new kid on the block in tumorigenesis with cancer-associated mycobiomes extracted from several cancer types. For PDAC, colonization with the yeast *Malassezia* seems to promote cancer progression, already in precursor lesions. One responsible mechanism appears to be complement activation via the lectin pathway. In the present article, we review the role of the complement system in tumorigenesis, presenting observations that propose it as the missing link between fungal dysbiosis and PDAC development. We also present the results of a small pilot study supporting the crucial interplay between the complement system and *Malassezia* colonization in PDAC pathogenesis.

**Keywords:** tumorigenesis; mycobioma; complement; pancreas; tumor marker; *Malassezia*; fungal dysbiosis

## 1. Introduction

Cancer-related deaths represent the second most common cause of death in the world. The outcomes for people diagnosed with cancer have changed dramatically over the last five decades, with an average 5-year survival for all cancers increasing from 50% in the seventies to almost 70% in the current decade (National Cancer Institute https://www.cancer.gov, accessed on 1 October 2022). However, this trend does not hold true for all cancer sites. Despite important developments in oncologic treatment strategies, the 5-year survival for people diagnosed with Pancreatic Ductal Adenocarcinoma (PDAC) is still around 11% (National Cancer Institute https://www.cancer.gov, accessed on 1 October 2022), making it the third-leading cause of cancer-related mortality. These data reflect the distribution of the stages at presentation, with more than half of the patients already presenting metastases and only 15% of patients presenting with a respectable disease at diagnosis [1]. In addition, the prognosis remains limited in resected patients undergoing (neo) adjuvant treatment, with a median survival rate ranging between 6 and 20% [2]. This situation mirrors the need to develop further strategies aimed at earlier diagnosis of this pathology and at the identification of new treatment targets. Several publications highlight a pivotal role of the complement system in PDAC tumorigenesis,

suggesting its components not only as potential serum markers [3,4], but also as crucial for progression towards a metastasized stage [5]. Paired with the recent knowledge about the role of mycobioma in tumor development [6,7], this review aims to give an overview on the complement system and its role in PDAC tumorigenesis including findings that suggest a link between fungal colonization of the tumor and complement activation in PDAC tumorigenesis. The final chapter presents the results of a small pilot study that supports the view to further pursue research efforts on this topic.

## 2. Materials and Methods

A systematic PubMed/MEDLINE literature search was performed for the role of complement in tumorigenesis, with a focus on PDAC. Keywords included "complement system", "tumorigenesis", "cancer" in combination with "pancreas" and "pancreatic ductal adenocarcinoma". With regard to the pilot study presented in chapter 6, see Appendix A for a detailed material and methods description.

## 3. The Complement System: Effector and Immune Hub

Tumors are more than only an aggregation of transformed cells. Instead, they represent a complex mass composed of multiple local and recruited cell types and soluble compounds [8]. A significant part of these elements belongs to the innate or adaptive immunity system [8,9]. Their activation is triggered by abnormalities of the malignant cells, such as altered cell morphology and generation of tumor-specific antigens that distinguish cancer cells from their non-transformed counterparts. The resulting inflammatory processes form a specific immunological tumor microenvironment (TME) that actively participates in multiple cancer hallmarks, such as cell proliferation and survival, tumor angiogenesis, metastasis, immune evasion, and resistance against therapies.

One important contributor to the TME is the complement system. Complement is an evolutionarily ancient, multifunctional system of innate immunity. It consists of close to sixty soluble and membrane-bound proteins including activating proteins, regulators, and receptor molecules. All of them act together to form a tightly regulated cascade of cleavage and assembly reactions, thereby generating effector molecules that fulfill various functions [10–12]. The main production site of complement factors is the liver, but the local synthesis of various factors is found in most organs, including the pancreas [12–16]. Figure 1 gives an overview of the complement cascade with its three activation pathways.

The classical pathway (CP) is initiated by the binding of complement factor C1q to antibodies/immune complexes on the surface of foreign target structures [17]; therefore, CP serves as an effector arm of the adaptive immune system and is referred to as "antibody-dependent". The lectin pathway (LP) shares several steps and involved molecules with the CP, but is triggered by sugar molecules commonly found on the surface of microbial cells. The main starter molecule of LP is mannan-binding lectin (MBL), but also ficolins, are able to initiate the same processes [18]. Both CP and LP result in cleavage of the factors C4 and C2 that form the C3 convertase C4b, 2a.

The alternative pathway (AP), often called 'tickover', is activated slowly and continuously in plasma by hydrolysis of factor C3. In the presence of foreign surfaces, a complex of C3b and the fragment Bb, derived from cleavage of factor B, is formed [19]. This C3b, Bb complex is stabilized by properdin (P) to generate the C3 convertase C3b, Bb, P of the alternative pathway [17]. In addition to spontaneous activation, the alternative pathway can propagate the complement cascade by entering the amplification loop [20].

Both C3 convertases efficiently perform the central step of the complement cascade, the cleavage of complement factor C3 into the fragments C3a and C3b. C3b is an efficient opsonin that marks foreign structures for phagocytosis, but also participates in the further proceeding of the complement cascade. It attaches to one of the two C3 convertases to form the respective C5 convertases. These enzymes catalyze the second central step of the complement system, the cleavage of C5 into C5a and C5b. C5a, together with C3a, represents an important pro-inflammatory and chemotactic mediator and affects multiple

processes in the body [21]. The fragment C5b initiates the terminal complement pathway that results in formation of the lytic C5b-9 complex, also termed membrane attack complex (MAC) or terminal complement complex (TCC). This complex forms a pore in the attacked surface, thus destabilizing the intracellular environment of pathogens or altered cells and resulting in their inevitable death.

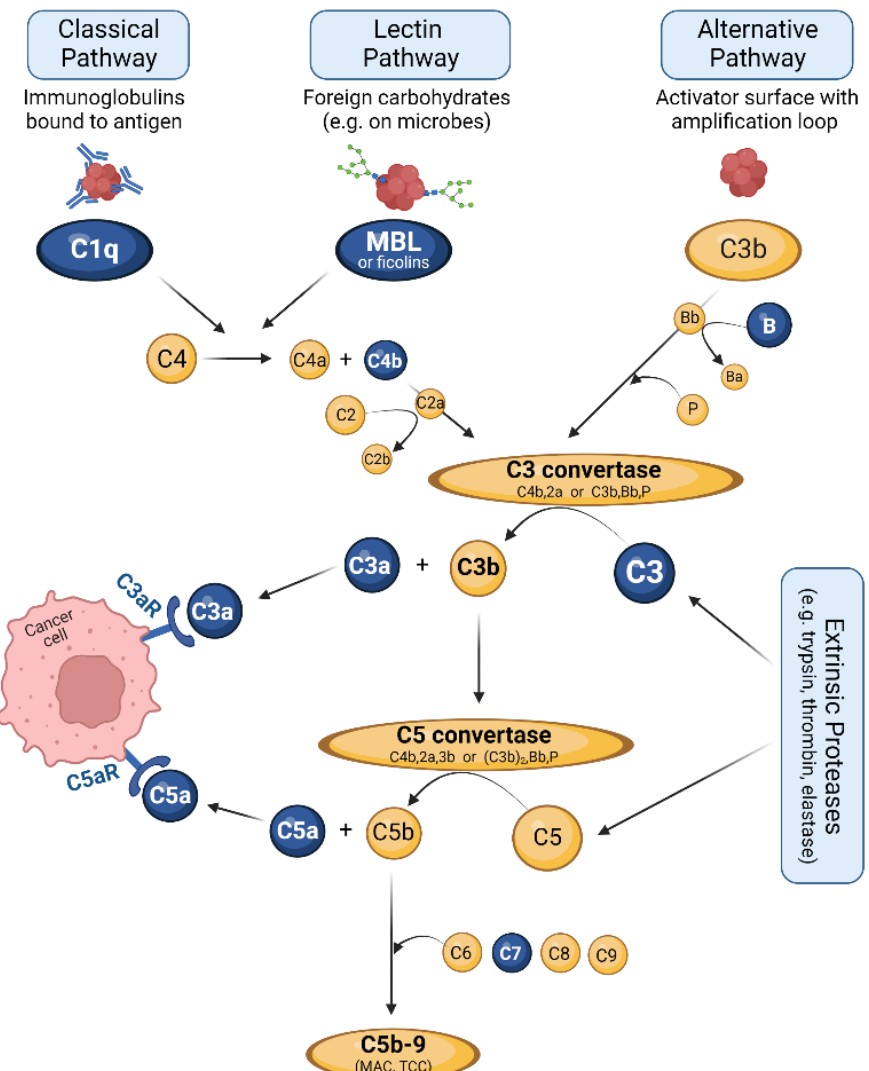

**Figure 1.** Overview of the three complement activation pathways and the subsequent cascade resulting in formation of the C5b-9 complex. Complement factors described as relevant for PDAC are labelled in blue. For further details, see text. MAC: membrane attack complex; TCC: terminal complement complex. Figure created with BioRender.com.

Apart from classical, lectin and alternative pathways, several bypass mechanisms also result in complement activation, such as those mediated by thrombin [22], cathepsin [23] or trypsin [24] (see also Figure 1). Since trypsin represents a central enzyme that is produced and released by pancreatic exocrine glands, the cleavage of C3 and C5 by this enzyme is of particular relevance in pancreas-associated conditions [25].

The role of the complement system is often focused on destruction of pathogens and includes their direct lysis via the C5b-9 complex and their opsonic tagging for phagocytosis; the latter is achieved via deposition of C3b and C3b-derived fragments on the pathogen surface, facilitating their recognition and engulfment by phagocytes [26]. Furthermore, complement promotes pro-inflammatory responses to control infection, mainly via the anaphylatoxins C3a and C5a [27,28]. However, the spectrum of complement-associated

functions is much broader than just antimicrobial attacks. Removal of apoptotic/dead cells as well as clearance of immune complexes from circulation to maintain and re-establish homeostasis are further core tasks of complement. Complement is also a key orchestrator of both innate and adaptive (B-cell and T-cell based) immunity and regulates their activity [27,28]. The multifaceted roles of complement in cancer are explicitly reviewed below and specified for pancreatic cancer.

By nature, the complement system attacks everything that is not actively protected by a plethora of regulator proteins that emit the message 'do not attack' [21]. To avoid collateral damage of self-tissues, very tight regulation and protection of host cells are of utmost importance. This is reflected by the considerable number of soluble or membrane-bound regulatory proteins that exceed the number of complement factors. Only a few examples relevant for PDAC are mentioned here; for a detailed overview, see [29]. Although complement regulators work on multiple levels of the cascade, most of them affect the generation or function of either C3 or C5 convertases. Common membrane-bound regulators protecting the vast majority of body cells from complement attack are CD46, CD55 and CD59. Whereas CD46 and CD55 promote C3 degradation or decay of C3 convertases, CD59 inhibits the assembly of the C5b-9 complex. The soluble regulators C1 inhibitor (C1-INH) and C4 binding protein (C4bp) both down-modulate the proceeding of classical and lectin pathway, C1-INH by blocking the relevant serine proteases and C4bp by promoting the decay of C3 convertase of these pathways. The soluble regulator factor H (fH) targets the alternative pathway and accelerates the decay of the respective C3 convertase [29].

## 4. Complement as Part of the TME

Cancer is much more than a mass of transformed cells that grow autonomously. The growing cell mass is sustained and protected by a tumor-associated stroma, composed of multiple infiltrated and local cell types as well as various proteins. This complex and diverse environment is referred to as tumor microenvironment (TME). The pivotal role of TME in cancer progression became obvious in the last years and is thoroughly reviewed elsewhere (e.g., [30,31]).

Besides extracellular matrix proteins (collagens, proteoglycans etc.) and signaling molecules such as cytokines and chemokines, the complement system is a main player of the TME protein components. Its role in modulation of tumor development and in orchestration of TME is reviewed and experimentally confirmed for PDAC in the following chapters (see below). The most relevant cell types for the role of complement in the TME of PDAC are (beside the cancer cells themselves) cells related to the innate immune system such as myeloid-associated cells such as tumor-associated macrophages (TAMs) and tumor-associated neutrophils (TANs), as well as cancer-associated fibroblasts (CAFs) and pancreatic stellate cells (PSCs) which are crucially involved in stroma formation.

TAMs represent a highly abundant immune cell type within tumors that are able to fulfil a broad repertoire of functions via diverse phenotypes [32]. TAMs can be categorized into M1 and M2 subsets that differ in their contribution to tumor development. The M1-like pro-inflammatory phenotype mediates a potent tumor-suppressive immune response, whereas M2-like TAMs contribute to tumor-supporting procedures such as angiogenesis, immunosuppression and metastasis. The TAMs that reside within the tumor generally show characteristics of the M2 phenotype [33].

Tumor-associated neutrophils (TANs) also show N1 or N2 phenotypes. After being recruited, neutrophils can be polarized by various cytokines or chemokines to one of the phenotypes and subsequently play an either tumor-suppressing or tumor-promoting role [34]. The antitumor N1 phenotype enhances cytotoxicity against tumor cells and attenuates the local immunosuppression, whereas the protumor N2-TANs participate, e.g., in metastasis development.

Pancreatic stellate cells (PSCs) are pluripotent mesenchymal cells that exist typically in a quiescent state. In a normal pancreas, they reside around acinar cells and contain large amounts of vitamin A-containing lipid droplets [35]. After being activated by a variety

of triggers, PSCs subsequently undergo a morphological change into myofibroblast-like cells and change their functionality spectrum. Nowadays, it becomes more and more clear that they represent critical players in pancreatic pathophysiology, pancreatitis and pancreatic cancer [35,36]. Activated PSCs harbor immunological functions, can recognize pathogen-associated molecular patterns and engulf pathogens [37]. Furthermore, activated PSCs highly upregulate the expression of stroma proteins and thus crucially contribute to the excessive fibrosis of aggressive PDAC. The synthesized desmoplastic stroma promotes the formation of a microenvironment that favors malignant transformation and facilitates the capacity of cancer cells to survive and invade [38].

Cancer-associated fibroblasts (CAFs) are derivatives from PSCs, but also from other cell types in the TME. They represent an important cellular component of the TME and are also main producers of the various stroma proteins [39]. CAFs are heterogeneous and can be divided in different subpopulations: myofibroblast CAFs (myCAF), inflammatory CAFs (iCAF), antigen-presenting CAFs (apCAF) and complement-secreting CAFs (csCAF) [39,40]. CAFs are involved in immune regulation of the TME by inhibiting CD8+ cells and producing immune-modulatory cytokines/chemokines. Via secretion of these molecules, CAFs are supposed to affect cancer cell proliferation and functionality of other immune cells (TAMs, TANs, PSCs) in the TME [39].

## 5. The Complement System in Tumorigenic Processes of PDAC

The neoplastic transformation of cells induces modifications of the glycosylation pattern and aberrant phospholipid metabolism; the subsequent alterations of the cell membrane target the transformed cells for recognition by complement ([41,42] reviewed in [9]). Multiple interactions between tumor and complement were described for various tumor types (summarized in Figure 2). As listed below, there are strong indications that the majority of these interactions might also be applicable for PDAC. Unfortunately, there is insufficient information in the overwhelming number of studies, where these processes occur, and this represents a critical point in the current state of knowledge. Future studies are urgently needed to address this paucity of data.

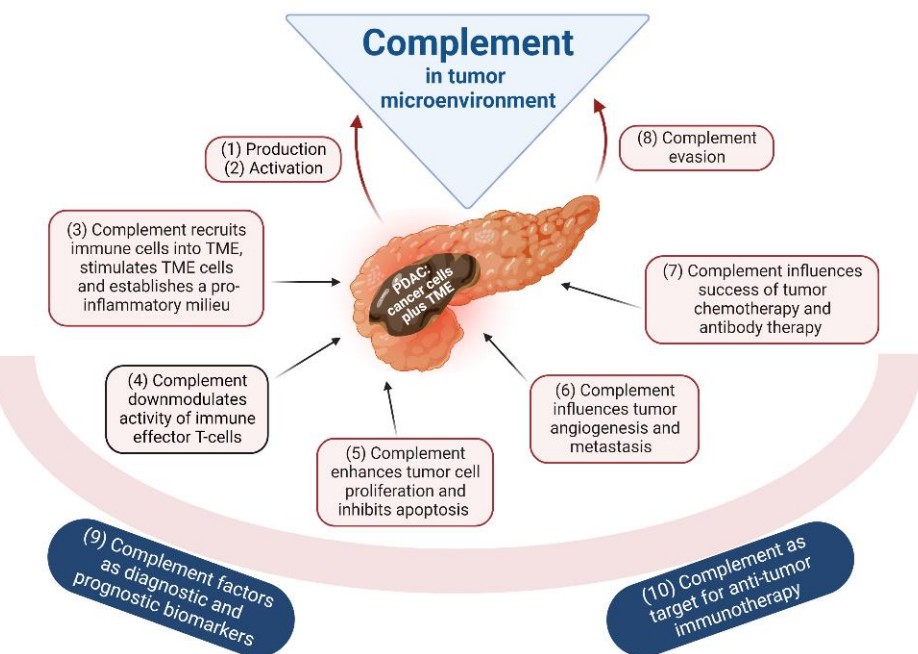

**Figure 2.** Putative encroachments of the complement system in the process of tumorigenesis, and, vice versa, putative interventions of tumors on complement in the TME (tumor microenvironment). Furthermore, putative applications of complement as PDAC biomarkers or in therapy are indicated. Numbering refers to the description in Section 5 of the text. Figure created with BioRender.com.

## 5.1. Complement Production in Tumors

Many tumor types are able to synthesize single complement factors and to express complement receptors [43,44]. Lu et al. [45] even reported for several tumor types a close correlation between the level of complement gene expression and clinical prognosis. In addition, recruited immune cells can contribute to the complement protein level in the tumor.

For PDAC, a distinct TAM subpopulation in the TME of PDAC patients was identified to upregulate the expression of the starter molecule of the classical pathway, C1q, even before the appearance of liver metastases [4]. This enhanced C1q synthesis can be correlated with the prognosis of PDAC patients [5,46], thereby underlining the relevance of complement for tumor progression. The expression of alternative pathway molecules is also stimulated, as shown by several-fold higher levels of factor B (fB) in transformed pancreas cells compared to normal human pancreatic duct epithelial cells [47].

Another central component of the PDAC-TME, the CAFs also contribute to the local complement production. As described above in Section 3, CAFs are a heterogenous celltype with different phenotypes. The recently identified subset csCAFs is designated for its capacity to produce complement system components [48]. Single-cell RNA sequencing of PDAC detected highly upregulated transcription of the factors C3, C7, fB, fD, fH and fI. This spectrum comprises both proteins that directly participate in the cascade and regulatory proteins that modulate complement activity. Because CAFs represent a considerable fraction of the cellular TME, their contribution to the local presence of complement might be substantial. In addition, complement production was demonstrated for specified subsets of TAMs and include C4b and C1q [40].

Furthermore, the expression of complement proteins C3 and C4b1 (a gene variant of C4) was shown to be significantly higher in PDAC tumor tissue than in normal pancreatic tissue without identification of the cellular source [3,49]. Interestingly, the C4b1 synthesis strictly correlated with TNM staging and might therefore allow the diagnosis of advanced PDAC. In contrast, no correlation was detected between expression levels of C3 with tumor TNM staging, indicating that its upregulation is a rather early process in PDAC tumorigenesis [49].

## 5.2. Complement Activation by the Tumor

As described above, membrane abnormalities of tumor cells trigger complement activation; many reports identified the classical pathway to be relevant, but also the lectin and alternative pathway were found to be involved ([50,51]; reviewed in [52]).

Until now, there are no precise analyses for the capacity of PDAC cells to stimulate the complement cascade. However, complement activation products such as C3a and C5a are found in the tumor environment and can be supposed to originate from recognition of the cancer cell surface by the corresponding pathway starter molecules. It is a tempting project to compare the complement cascade induction by PDAC versus normal pancreatic cells in more detail.

Not only tumor cells themselves but also TME components of PDAC can activate complement and thus modulate the inflammatory milieu. Single-cell resolution identified apCAFs to harbor complement activation functions, although the precise mechanism remained unclear [40,53]. The same study revealed complement activation also by TAMs [40].

## 5.3. Complement-Mediated Inflammation, TME Cell Recruitment and Activation

Complement is a main driver of the recruitment of lymphoid and myeloid cells into tumors and their TME [45,54]. These cells, particularly TAMs and TANs, participate in the establishment of the pro-inflammatory microenvironment, but also interfere with an efficient anti-tumor immunity (see Section 5.4; [31]). The complement anaphylatoxins C3a and C5a are potent mediators of macrophage and neutrophil recruitment [55,56]. Furthermore, C3a and C5a are efficient stimulators of inflammatory mediators like IL-

1β, IL-6 and TNF-α [55], and inflammatory processes are vital for tumor development, promoting subthreshold neoplastic tumor states to fully cancerous states [8,57,58].

For PDAC, complement-mediated recruitment can be supposed by a positive correlation of the presence of TAMs and other immune cells with a high stromal factor B (fB) expression [59]. Complement-driven activation of local immune cells in the pancreatic TME is described for PSCs. In vitro and in vivo studies demonstrate that the anaphylatoxin C5a is capable to stimulate PSC and thus to enhance the production of extracellular matrix proteins [60]. Additionally, the TANs in the TME can be supposed to be activated by complement, since this cell type harbors a variety of complement receptors on its surface [61].

### 5.4. Complement and Local Immunosuppression in the Tumor

Tumors can reduce the functionality of T-cells against tumor-specific antigens. This downmodulation of immune cell activity is mainly mediated by cells of lymphoid and myeloid origin (e.g., TAMs) in the TME [62]; however, complement activation products and regulators orchestrate this immune suppression in many ways (reviewed in [9]).

For PDAC, only few studies exist that investigate the contribution of complement to immunosuppression in this type of tumor. Shimazaki et al. [59] could correlate the stromal factor B (fB) expression in PDAC TME with the enrichment of myeloid-derived suppressor cells and with immunosuppressive regulatory T-cells. The complement regulator C4-binding protein (C4bp) acts positively on the tumor immunology by directly activating B-cells and favoring the accumulation of tumor-infiltrating lymphocytes [63].

### 5.5. Complement and Tumor Growth

Complement factors are able to stimulate cancer cell proliferation and to interfere with their apoptosis [64]; these effects were shown, e.g., for the anaphylatoxins C3a and C5a [9,31,52,65]. In addition, complement proteins regulate the activity of TAMs, an important component of the TME, that non-specifically promote the growth of the tumor [43].

For PDAC, a tumor-promoting activity was found for complement factor B (fB). As demonstrated with pancreas cancer cells, downmodulation or overexpression of fB influences tumorigenesis [47,59], with induction of the PI3-AKT signaling pathway as a likely molecular mechanism. Similarly, the activation of PSC by complement C5a with subsequent upregulation of ECM protein production [60] can be supposed to favor tumor growth. TANs, another complement-inducible cell type of the TME, are also important promoters of PDAC growth [66,67].

Aykut et al. [68] established a further process by which complement can promote PDAC progression. Using tissue samples from PDAC patients as well as a PDAC murine model, the group found an about 3000-fold enrichment of fungi in PDAC compared to normal pancreatic tissue. The fungus-enriched pancreatic oncobiome was supposed to originate from migration, mainly of the yeast *Malassezia globosa*, from gut lumen into the pancreas. The presence of *Malassezia* in the pancreas induced chronic activation of the complement system via the lectin pathway. Deletion of its starter molecule MBL significantly interfered with tumor progression in a mouse model; in addition, PDAC patients with high MBL expression had a worse prognosis compared to patients that synthesized low MBL amounts. Complement activation resulted in the generation of the pro-inflammatory molecule C3a. Binding of this anaphylatoxin to its receptor on the tumor cells can trigger tumor promotion, as shown in animal experiments with application of recombinant C3a into the PDAC tissue [68,69].

### 5.6. Complement in Tumor Angiogenesis and Metastasis

Most studies described a pro-angiogenic effect of complement factors (mainly C1q and C5a) in tumors, but occasional papers also report an anti-angiogenic impact of complement for some tumor types [9,43,70,71]. Complement proteins have also been described to participate in metastasis by influencing some of the key steps of this process, such

as epithelial-to-mesenchymal transition (EMT) of cancer cells or the degradation of an extracellular matrix [9,45].

PDAC-derived metastases are detected in up to 50% of patients at the time of first diagnosis, with a remarkable predilection for the liver [72]. This process was recently studied in detail, showing that PDAC-derived exosomes promote the formation of a fibrotic environment in the liver and thus establish a pre-metastatic niche. A crucial trigger for this preparation of PDAC metastasis in the liver is the delivery of complexes between complement factor C1qbp (C1q binding protein) and CD44v6 via the PDAC-exosomes [73]. A further study confirmed the role of complement, showing that upregulation of C1q expression in tumor-associated macrophages contributes to tumor metastasis [5]. In vitro experiments with PDAC cell lines showed that exogenous human C1q could promote cell migration and invasion, thus establishing a crucial role for this complement molecule.

The EMT of PDAC cells was linked with the complement factor C3-derived anaphylatoxin C3a and its receptor C3aR. EMT comprises a bundle of PDAC cell changes, which collectively translate into enhanced migration and metastasis [74]. In vitro studies showed that C3a activates PDAC cell signalling pathways that result in these EMT processes. Furthermore, C3aR tissue quantifications revealed its increased expression in patients with metastatic lesions [75]. Moreover, the C5a anaphylatoxin can stimulate EMT, and histological examinations detected its receptor C5aR in liver metastases [76].

The promotion of PDAC metastasis by complement might also be indirect via activation of the TANs. Neutrophils in the PDAC TME were shown to significantly enhance migration of pancreatic cancer cells, and neutrophil depletion could suppress tumor growth and metastasis (reviewed in [67]).

*5.7. Complement and Efficacy of Anticancer Therapy*

Complement factors upregulate inflammatory mediators (cytokines, chemokines), which in turn can stimulate chemo-resistance pathways [58,65] and thus limit the success of chemotherapy. On the other hand, the response on anti-tumor therapy with monoclonal antibodies can be enhanced by complement since these antibodies, bound to tumor-specific antigens on the cell surface, activating the classical complement pathway. A consequence is improved tumor cell elimination via antibody-dependent cell-mediated cytotoxicity (ADCC), complement-dependent cytotoxicity (CDC) and complement-dependent phagocytosis (CDP) [77].

Drug resistance is of particular relevance for PDAC since this is one of the major reasons for high mortality. A recent article reviews the different signaling pathways that play a central role in causing chemo-resistance in pancreatic cancer. The anaphylatoxins C3a and C5a are trigger molecules for all of them via upregulation of relevant inflammatory mediators like TNF-, IL-1$\beta$ and IL-6 [58].

*5.8. Complement Evasion of Cancer Cells*

Most cancer cells developed strategies to avoid and resist the destructive attack of complement [78]. These strategies not only interfere with the direct lysis of the transformed cells, but also affect the therapeutic success by monoclonal antibodies. A common evasion mechanism of cancer cells is the high expression of membrane-bound or the acquisition of soluble complement regulators on the surface [65]. Alternatively, tumor-derived proteases might eliminate complement proteins by cleavage, or endocytotic processes can remove the lytic C5b-9 complex from the surface [79,80].

Unfortunately, only limited studies exist for complement evasion by PDAC. One described mechanism is the over-expression of membrane-bound complement inhibitors like CD46, CD55 and CD59 by PDAC cell lines compared to non-cancer pancreatic cells [81,82]. Pro-inflammatory signals further increase the expression of these complement inhibitors [83].

Another relevant evasion strategy is based on the C1 inhibitor (C1-INH) which inactivates the proteases C1r, C1s and MASPs in the complement system. Elevated C1-INH concentrations were measured in the plasma obtained from pancreatic cancer patients [84]

and also in PDAC tumor tissue, compared to normal pancreatic tissue samples [85]. The clinical relevance of C1-INH for complement evasion of the tumor is underlined by the fact that PDAC patients with high C1-INH expression had a 5-year survival of 15% compared to 56% when the C1-INH level is low [85].

Complement evasion might be relevant not only for the pancreas tumor cells themselves but also for pro-tumorigenic actors like the above-mentioned *Malassezia globosa* (see Section 5.5). An interesting project might aim to quantify whether *Malassezia globosa* is able to escape from complement attack and thus establish itself as a tumor-promoting trigger for PDAC progression.

### 5.9. Complement as Tumor Biomarker

The above-mentioned bundle of interventions in the process of tumorigenesis makes complement an interesting marker for both; tumor diagnosis to shorten the time until onset of therapy and for tumor prognosis to choose and design the optimal therapy [45]. The currently-used PDAC biomarkers CEA (carcinoembryonic antigen) and CA19-9 (carbohydrate antigen 19-9) lack satisfactory specificity and/or sensitivity. Therefore, there is considerable effort either to replace them with more applicable ones or at least to supplement them to get a diagnostically conclusive panel.

In this respect, several complement proteins have been proposed as biomarkers to ameliorate PDAC diagnostics [86]. An interesting candidate is C1q, which is highly expressed by a subset of tumor-associated macrophages. This cell population can be detected not only in the tissue of primary PDAC tumors and their liver metastases but also in the patient's blood, allowing an easy quantification of C1q as a disease marker [4].

Further promising candidates are C3 and its derivative, the anaphylatoxin C3a. Considerably higher serum levels of both molecules were measured in PDAC patients compared to non-tumor control individuals, proposing them as potential tumor markers [3,75].

One of the most promising and extensively studied marker candidates is complement factor B (fB). Lee et al. [47] described two-fold higher levels of fB in plasma samples of PDAC patients than in samples derived from healthy individuals. A comparison of fB with the established PDAC marker CA19-9 revealed a higher specificity for pancreatic cancer than for other tumor types and better discrimination between normal and transformed pancreatic tissue; an additional improvement in PDAC diagnostics could be achieved by combining those two markers [47]. A complementary study proved fB to be one of the most robust upregulated proteins in PDAC [59]. Complement fB might also be useful to predict the outcome of PDAC; patients with high fB levels in serum exhibit a shorter disease-free survival than patients showing low concentrations of fB [87].

A special position under all putative marker proteins can be attributed to complement factor C7. C7 might serve as a meaningful and valuable prognostic biomarker for PDAC-associated cachexia, a powerful predictor of mortality in PDAC. C7 expression levels in plasma excellently correlated with the cancer weight-loss grade, and the relationship was tightest under all detected proteins whose expression correlated with the percentage of weight loss in PDAC patients [88].

Whereas the majority of PDAC biomarker candidates were studied in blood samples, complement quantification might also be interesting in pancreatic biopsies. The studies of Chen et al. [3,49] imply that C3 upregulation in the tissue occurs early in tumorigenesis and thus allows diagnosis of early-stage PDAC. In contrast, enhanced C4b1 is stage-dependent and, therefore, useful as a diagnostic marker for advanced PDAC. Further studies demonstrated the reliability of fB also as a tissue marker for PDAC prognosis, since high stromal fB concentrations correlated with unfavorable clinical outcome and an increased risk for dissemination after surgery [59].

*5.10. Complement as a Therapeutic Target*

The versatile roles of complement in carcinogenesis imply finally that this innate immune system represents a promising candidate for therapeutic interventions. Modifying the complement activity can eliminate all its above-mentioned tumor-promoting effects, e.g., the induction of inflammatory processes, tumor growth and metastasis, suppression of T-cell-based antitumor immune response, and chemo-resistance. A further complement-based strategy is the optimization of therapeutic monoclonal antibodies in order to achieve maximal complement activation and tumor cell attack via C5b-9 complex. This perspective is particularly promising in PDAC, with limited therapeutic success until now. Multiple complement factors represent interesting targets for immunotherapy, e.g., the inhibition of C1q with its crucial pro-metastatic properties [5]. Further studies are needed to identify the most promising strategy, targeting either single or bundles of complement proteins.

## 6. Fungal Dysbiosis and PDAC Development

In recent years, more and more attention was pointed at the role of the microbial flora in the initiation and progression of pancreatic cancer. Interestingly, the role of the fungal flora, so called mycobiota, arose as a key factor in tumorigenesis, and a distinct mycobiome signature was demonstrated to characterize several types of cancers [7].

Not only with regard to tumors of the hollow organs of the gastrointestinal tract, like colorectal [89] and esophageal cancer [90], specific fungal signatures have been described. Additionally, within the human pancreas, an organ formerly thought to be sterile, there is increasing evidence of the presence of a specific mycobioma [91,92]. Initially described in 2019 by Aykut et al. and recently confirmed by other independent studies, 18S rRNA sequencing and immunohistochemistry analysis revealed a change in the human intrapancreatic mycobioma not only in the presence of PDAC but also in the presence of precursor lesions like PanIN [7,68,93]. However, all these techniques demonstrate only the presence of fungal components like nucleic acid or membrane compounds like beta-glucan, and up to now, any attempt to cultivate living fungi from tumor specimens has been unsuccessful [7].

Further metagenomic characterization of the mycobiome showed absolute abundance of *Malassezia* spp. within human and mouse PDAC [7,68] and also of *Alternaria* spp. [7]. *Malassezia* is a known skin commensal with capability of gut colonization in humans [94], and its presence within the pancreatic gland is thought to occur through a direct migration from the duodenum via the main pancreatic duct [68]. Of note, *Malassezia* spp. encodes some secreted enzymes—similar to those of *Candida* spp.—that have been shown to act carcinogenically [95,96].

From a clinical point of view, fungal dysbiosis was demonstrated to be related to the tumor burden. The ablation of the mycobiota resulted in prolonged survival and reduction of tumor progression in an orthotopic mouse model for PDAC. Interestingly, only the species *Malassezia* showed the capability of promoting tumor growth in PDAC, while other fungi like *Aspergillus* spp. or *Candida* did not [68].

Based on these findings, different pathogenetic models have been proposed to understand the pathways involved in this fungal-driven tumorigenesis [92]. Principally, the link between the immune milieu within the tumor and its response to fungal components are thought to represent the crucial aspect of this process. Alam et al. showed that increased immune-suppressive cell populations like Th2 and innate lymphoid cells 2 (ILC2)—which characterize the tumor microenvironment (TME) of PDAC and can lead to its progression—are recruited by IL-33 secreted by PDAC cells. Of note, the secretion of IL-33 was demonstrated to be dependent on the composition of the intratumoral mycobioma [93]. So far, the link between the fungal presence and the extracellular release of IL-33 has not yet been defined.

Aykut et al. observed a crucial role of the complement system in relation to the tumorigenic processes within the pancreas. More specifically, the model proposed by this study group implies the recognition of *Malassezia* ssp. by MBL and the activation of the lectin pathway of the complement with consequent production of C3. These findings are

corroborated by the positive correlation between MBL and C3 with poor survival in PDAC patients [68].

These two independent studies showed interesting links between the intrapancreatic mycobiota and the interdependent pro-tumorigenic immunity shape. However, many questions remain open. It is still unclear which receptors and specific mechanisms can shape the TME in such a pro-tumorigenic way. Furthermore, no specific tissue-related analysis concerning the expression of complement factor within pancreatic tissue was performed. It is still unclear whether the tumor cells themselves can produce complement factors or activate their cascade.

Curiously, the role of the reactive species of the oxygen (ROS) in response to intra-pancreatic fungal colonization remains quite unexplored. Still, these compounds could reveal new interactions between the different tumorigenic pathways. On the one hand, ROS can activate extracellular release of IL-33 [97,98]. On the other hand, lectin comple-ment pathway activation can occur in a context of oxidative stress [99]. Moreover, IL-33 and complement anaphylatoxins are known to interact on mast cells [100] synergistically. Whether this synergism happens on other immune cells, or even on PDAC cells themselves, is an intriguing question.

Strong evidence suggests a link between fungal dysbiosis and immunologic alterations resulting in a pro-tumorigenic milieu within the pancreas. With the complement system being crucially involved in these mechanisms, analyzing the expression of the complement factors within pancreatic tissue is of primary importance to further understand its role in tumor development.

## 7. Pilot Study: Complement and Malassezia in PDAC

As illustrated above, new insights underline the relevance of complement in the context of PDAC development. Furthermore, the model of fungus-driven complement activation (described in Section 6) corroborates the role of complement as a main orchestra-tor in the TME. However, additional aspects need to be further clarified. In the study of Aykut et al. [68], the focus is set on the role of the lectin pathway, but no consideration has been made in regard to the classical and alternative pathway of complement. Moreover, the relation between the colonization by *Malassezia* and the effective level of complement expression and/or secretion remain still unknown.

Intending to further characterize this interplay between the complement system and fungal dysbiosis, we set up a pilot study based on the retrospective analysis of formalin-fixed paraffin-embedded (FFPE) specimens of pancreatic lesions. Following approval from the ethics committee of the Medical University of Innsbruck (study number 1188/2021), 19 pancreatic lesions, resected between 2017 and 2019 at the Department of Visceral, Trans-plant and Thoracic Surgery of the Medical University of Innsbruck, were included in the study.

The intent of the pilot study consisted in analyzing the specimens (1) for the presence of DNA fragments of *Malassezia* spp., which had been recently described to be involved in PDAC progression [68], and (2) for the expression of selected relevant complement proteins. More precisely, our examination focused on C1q as starter molecule of the classical pathway, on factor B as specific for the alternative pathway, and on MBL as a pattern recognition molecule of the lectin pathway. In addition, we analyzed the presence of C3, the central complement factor of all three pathways, acting as an opsonin and as starter molecule of the terminal pathway. From the histopathological point of view, we focused on entities arising from the exocrine component of the pancreas, including PDAC, as well as premalignant and benign lesions. Stroma, epithelium and inflammatory cells of the lesions were evaluated separately. For all 19 cases, lesional as well as perilesional specimens were available.

Following DNA extraction from each FFPE specimen, specific multiplex PCR using both primers for the ITS region of *Malassezia* spp. and species-specific primers for *M. globosa* and *M. restricta* were performed. The expression analysis of the different complement

proteins was done by immunohistochemistry (see Appendix A for a detailed Material and Methods description).

Among the 19 examined specimens, 10 were PDAC, and 9 were benign or premalignant pathologies, including 5 intraductal papillary mucinous neoplasms (IPMN), 2 cases of chronic pancreatitis, 1 pseudotumor in autoimmune pancreatitis and 1 serous cystic neoplasm (5.3%). The patients' demographic data are reported in Table 1.

**Table 1.** Patients' data of the specimens included in this study.

| Characteristics | *n* (%) |
|---|---|
| Sex ratio (M:F) | 11:8 |
| BMI * | 25.6 (16–43) |
| Diabetes mellitus | |
| Type I | 1 (5.3) |
| Type II | 3 (15.8) |
| Tobacco | 7 (36.8) |
| Alcohol | 7 (36.8) |
| Serum Bilirubin (mg/dL) * | 0.79 (0.0–13.0) |
| Histologic Diagnosis | |
| PDAC | 10 (52.6) |
| IPMN | 5 (26.3) |
| Chronic Pancreatitis | 2 (10.5) |
| Autoimmune Pancreatitis | 1 (5.3) |
| SCN | 1 (5.3) |
| Biliary Drainage | |
| ERCP | 5 (26.3) |
| With Stenting | 3 (15.8) |
| Preoperative Biopsy | 6 (31.6) |

* Median value (range); SCN: serous cystic neoplasm; ERCP: endoscopic retrograde cholangiopancreatography.

With regard to the fungal DNA, out of 19 specimens, positive *Malassezia* DNA isolation occurred in 14 cases. *Malassezia* DNA was present within the lesion in six cases, and in eight cases in the perilesional area. No specimen showed positivity for *Malassezia* DNA, both within the lesion and in the perilesional area simultaneously. Considering the distribution in relation to the histopathologic diagnoses, all 10 PDAC were positive for *Malassezia*, either in the lesion or in the perilesional area, while only four non-malignant lesions showed positivity for *Malassezia* ($p = 0.006$). Interestingly, three of these four non-malignant specimens containing fungal DNA were lesions known to harbor a higher risk of malignant transformation, namely one pseudotumor in autoimmune pancreatitis, one IPMN, and one chronic pancreatitis. No correlation was observed between the presence of *Malassezia* DNA and known risk factors for PDAC (age: $p = 0.354$, BMI: $p = 0.853$, diabetes: $p = 0.144$, tobacco: $p = 0.865$, and alcohol consumption $p = 0.865$).

The expression of each complement protein was analyzed separately in three different components of the pancreatic lesions, namely the tumor stroma, the tumor epithelium and the cellular infiltrates. All complement proteins analyzed (C1q, fB, MBL and C3c) revealed correlations either with the dignity of the lesion or with the presence of *Malassezia* DNA or with both.

Tumor epithelial cells displayed significantly higher C1q, C3c and MBL expression than benign/premalignant lesions ($p = 0.027$, $p = 0.034$, and $p = 0.027$, respectively; Figures 3 and 4).

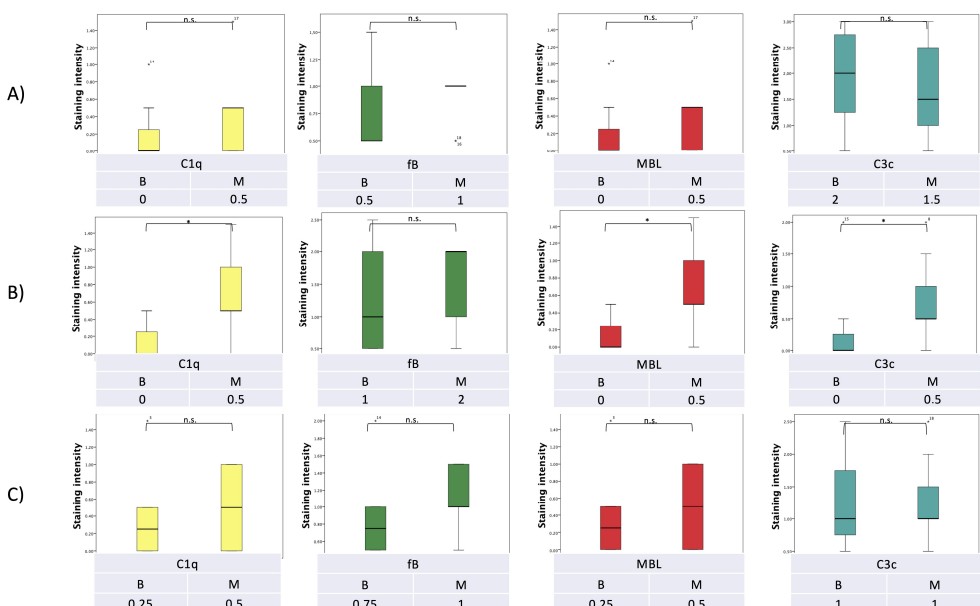

**Figure 3.** Semi-quantitative expression of complement factors (**A**) in the lesional stroma, (**B**) in the lesional epithelium, and (**C**) in the lesional inflammatory cells related to the benign (**B**) or malignant (**M**) dignity of the lesion (* *p* < 0.05; n.s. not significant).

In a similar pattern, a significantly higher expression of C1q, fB and MBL was detected in epithelial cells of lesions positive for fungal DNA compared to those not harboring fungal DNA (*p* = 0.003, *p* = 0.048 and *p* = 0.003, respectively), with C3c reaching almost statistical significance (*p* = 0.053).

With regard to the tumor stroma and the inflammatory component of the lesions, expression of C1q and MBL was significantly higher in *Malassezia* DNA-positive lesions compared to DNA-negative lesions (*p* = 0.041, *p* = 0.041, and *p* = 0.046, *p* = 0.046, respectively; Figures 5 and 6).

The low number of analyzed specimens in this pilot study must be considered when interpreting these data. Still, the important correlation between fungal DNA and the observed malignant pathologies confirms earlier findings regarding a putative tumorigenic role of *Malassezia* spp. It is noteworthy to highlight that not only all malignant specimens but also three premalignant lesions were found positive for fungal DNA. This suggests that the *Malassezia* spp. plays a pivotal role in tumor progression, an observation already made by the group of Aykut, who described in a mouse model the presence of *Malassezia* spp. as crucial for tumor development [68].

The results presented here suggest an intriguing interaction between the mycobioma and the complement system. In this regard, Aykut et al. already observed in PDAC activation of the complement system via the lectin pathway in the presence of *Malassezia*. As reported in Section 5.5, the deletion of MBL resulted in significantly lower tumor mass in a PDAC mouse model [68]. The present pilot study corroborates this observation by showing a correlation between MBL and *Malassezia* DNA in PDAC lesions. However, it also points at a possible involvement of other complement pathways.

Since not only MBL, but also C1q and C3c expression in epithelial cells correlated with *Malassezia* positive malignant lesions, activation of more than one complement pathway as pro-tumorigenic cause of fungal dysbiosis has to be taken into consideration. In fact, these findings are in accordance with previous reports of tumor cells [44] or recruited immune cells like TMAs [4] or CAFs [48], showing higher expression and production of different complement factors including not only MBL but also C1q and C3 (see Section 5.1). Similarly, these three complement factors have also been observed to be activated in tumoral microenvironment ([40,53], see Section 5.2) In addition, the higher expression of complement factors C1q and C3 found in our prelimianry study is in line with previous findings addressing

the complement as promising a sensitive and specific biomarker for PDAC (see Section 5.9). More precisely, the elevated presence of C1q and C3 in the serum of PDAC patients [3,4,75] correlated with the higher expression of the same factors within the pancreatic tissue. So far, there is still no evidence for the clinical use of serum MBL as possible marker for PDAC. However, in the light of our preliminary findings a characterization of MBL levels as well as of other complement factors in patients' serum could be an appealing project aiming at testing the specificity and sensitivity of these molecules in the PDAC setting but also at stratifying patients according to their tumor prognosis.

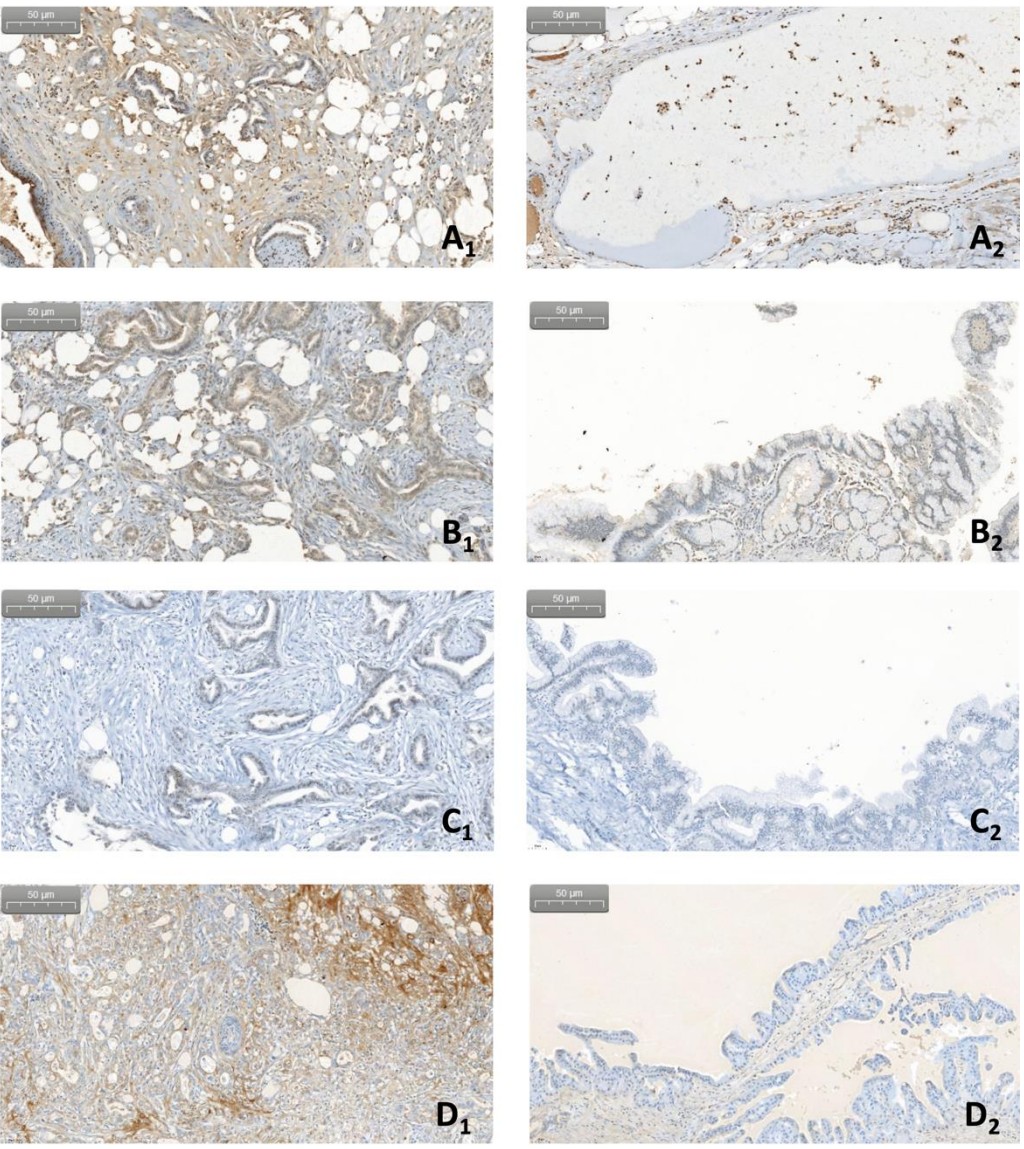

**Figure 4.** Immunohistochemistry. Comparison of the expression of complement factors between PDAC specimens (letters labelled with number 1) and specimens of benign or premalignant lesions (letters labelled with number 2: **A$_2$** corresponding to serous cystic neoplasm, while **B$_2$**, **C$_2$** and **D$_2$** depict intraductal papillary mucinous neoplasms, respectively). (**A**) expression of C1q; (**B**) expression of fB; (**C**) expression of MBL and (**D**) expression of C3c (magnification 20×).

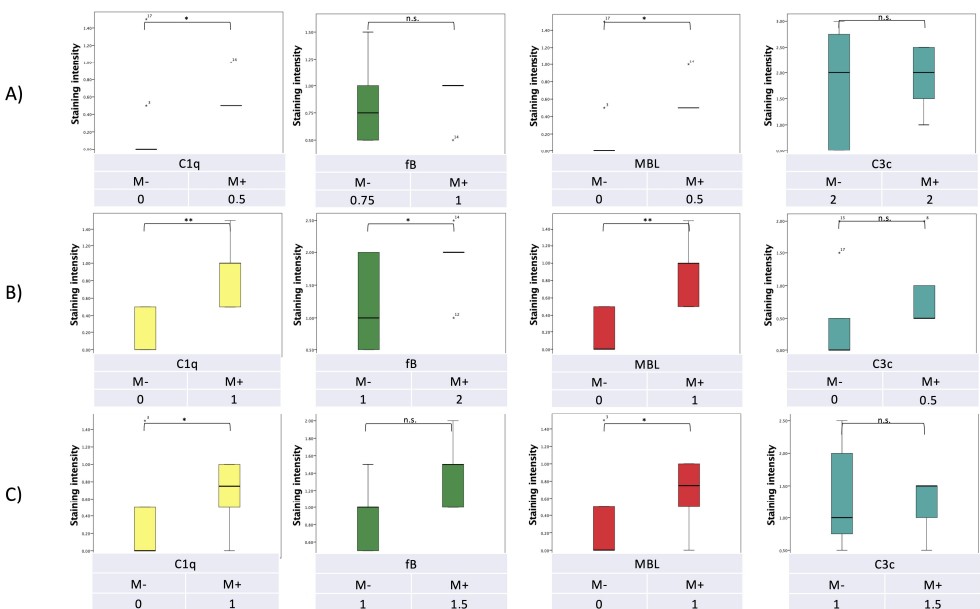

**Figure 5.** Semi-quantitative expression of complement factors (**A**) in the lesional stroma, (**B**) in the lesional epithelium, and (**C**) in the lesional inflammatory cells related to the presence (M+) or absence (M−) *Malassezia* DNA (* $p < 0.05$; ** $p < 0.005$; n.s. not significant).

In contrast, despite the observed correlation with the presence of fungal DNA in epithelial cells, there was no difference in fB expression between malignant and benign or premalignant samples. This is in conflict with the observation of fB as a highly-specific serum marker for PDAC [47]. Not having access to patients' blood samples in this retrospective study, we cannot exclude significant differences in the serum of these patients. Interestingly, our analysis revealed an increased expression of fB in the stroma of *Malassezia* ssp. positive lesions, although not reaching statistical significance. Even though not directly confirming the observations of Shimazaki et al., who described a high stromal fB expression in PDAC stroma with unfavorable clinical outcomes [59], this finding could point to an interplay between fB and *Malassezia* ssp. as a promoting factor for further tumor development.

The separate analysis of three distinct components of the specimen (epithelial cells, stroma, and inflammatory infiltrates) showed not only a correlation between complement factor expression and fungal DNA in the tumor epithelial cells, but also in the cells of the stromal and of the inflammatory compartment of the analyzed lesions. This confirms recent findings that complement activation does not only occur extracellularly when recognizing membrane abnormalities (see Section 5.2). In fact, in human T-cells, intracellular stores of complement proteins have been shown to be responsible to guide T-cell differentiation towards different subsets [101]. This suggests that the activation of the complement cascade could happen intracellularly in the epithelial pancreatic compartment, like in colorectal cancer, where intracellular complement participates actively in the tumorigenic process [102]. So far, mechanisms underlying its intracellular activation and regulation and its functional outcomes, in particular concerning the pancreas, are largely unexplored [36–45]. However, this aspect opens a new scenario regarding innate immune surveillance in relation to PDAC development.

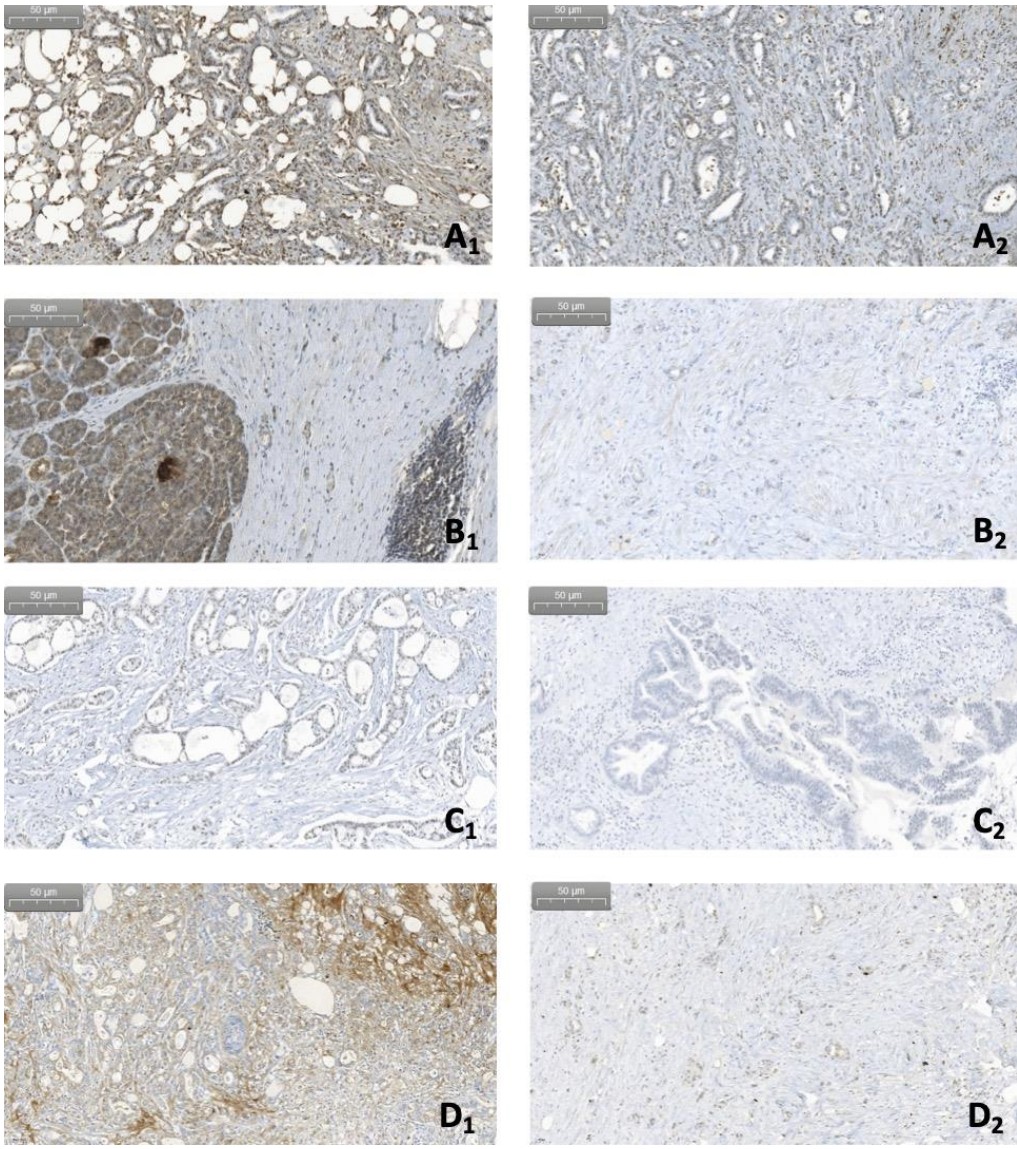

**Figure 6.** Immunohistochemistry. Comparison of the expression of different complement factors between specimens positive for *Malassezia* spp. DNA (letters labelled with number 1: $A_1$, $C_1$ and $D_1$ correspond to PDAC samples, $B_1$ to chronic pancreatitis) and specimens where *Malassezia* spp. was not detected (letters labelled with number 2: $A_2$, $B_2$, $D_2$ depict PDACs and $C_2$ corresponding to intraductal papillary mucinous neoplasm, respectively). (**A**) expression of C1q; (**B**) expression of fB; (**C**) expression of MBL; and (**D**) expression of C3c (magnification 20×).

## 8. Conclusions

The complement system plays, as a part of the innate immune system, an important role in the TME of PDAC. Either directly on cancer cells or indirectly by promoting an immunosuppressive environment, it has been described to promote tumor development. An increasing amount of data suggests fungal dysbiosis as a pivotal tumor-promoter. In a mouse model, the activation of the lectin pathway hereby appears to be crucial. However, despite recent findings of cancer-associated mycobiomes in multiple cancer types, extracted, e.g., from The Cancer Genome Atlas (TCGA), the question whether fungal dysbiosis is correlated or causally associated with tumor development has not yet been answered [6,7]. An appealing suggestion that might path the way for new diagnostic and treatment targets is an interplay between the observed fungi and the complement system, resulting in a more tumor-permissive microenvironment.

The presented pilot study is consistent with previous publications, suggesting a link between fungal dysbiosis, the complement system and cancer development. An analysis of larger patient cohorts will reveal whether research efforts should continue in this direction.

**Author Contributions:** Conceptualization: C.S., R.B. and M.M.; methodology: C.S., R.B. and M.M.; formal analysis: C.S., R.B., G.S., G.R., B.T. and G.C.T.; investigation: G.S., B.T. and G.C.T.; data curation: R.B.; writing—original draft preparation: C.S., R.B., G.R. and M.M.; writing—review and editing: C.S., R.B., G.R., M.M, D.Ö. and C.L.-F.; visualization: G.S.; project administration: C.S. and M.M.; funding acquisition: M.M., C.S. and D.Ö. All authors have read and agreed to the published version of the manuscript.

**Funding:** This research was funded by the "In memoriam Gabriel Salzner Privatstiftung" to M.M., C.S. and D.Ö.

**Institutional Review Board Statement:** The study was conducted in accordance with the Declaration of Helsinki, and approved by the Ethics Committee of the Medical University of Innsbruck (1188/2021; approved on the 17 September 2021).

**Informed Consent Statement:** Informed consent was obtained from all subjects involved in the study.

**Data Availability Statement:** The data presented in this study are available on request from the corresponding author. The data are not publicly available due to privacy reasons.

**Acknowledgments:** We are indebted to Margot Haun and Magdalena Neurauter for expert technical assistance.

**Conflicts of Interest:** The authors declare no conflict of interest.

## Appendix A

*Appendix A.1. Materials and Methods*

Appendix A.1.1. Patient/Specimen Selection

Following approval from the ethics committee of the Medical University of Innsbruck (study number 1188/2021 approved on the 17 September 2021), a retrospective analysis of randomly-selected 19 malignant or benign and premalignant specimens of pancreatic lesions, resected between 2017 and 2019 at the Department of Visceral, Transplant and Thoracic Surgery, Medical University of Innsbruck, Austria, was performed. Patients' data were collected from a prospectively recorded and auditable medical database.

Appendix A.1.2. DNA-Extraction

The DNA extraction from pancreatic FFPE samples was performed with the QIAamp DNA FFPE Tissue Kit (Qiagen, Hilden, Germany), a well-established method for purification of genomic DNA from FFPE tissues. The procedure was carried out according to the manufacturer's instructions.

DNA extraction was performed on FFPE sections with a thickness of 5 μm prepared on TOMO adhesion microscope slides (Matsunami, Washington, DC, USA).

The sections were treated with xylol to remove paraffin. Further on, the samples were lysed under denaturing conditions with proteinase K for several hours. After a heat incubation at 90 °C for one hour to reverse formalin crosslinking, the DNA bound to the membrane in the extraction column; residual contaminants were washed away, and the pure, concentrated DNA was eluted from the membrane. A NanoDrop Spectrophotometer measurement determined the quantity and purity of the eluted DNA.

DNA samples were stored at −20 °C.

Appendix A.1.3. Multiplex PCR

To detect different *Malassezia* Species, a multiplex PCR was performed as previously described by Emre Vuran et al. [103].

We focused on two *Malassezia* strains: *M. globosa* and *M. restricta*, respectively. DNA amplification was carried out in a total volume of 50 µL containing isolated DNA, 1xPCR buffer with 1.5 mmol/L MgCl$_2$, 100 µmol/L each of dNTPs, 0.5 µmol/L of each oligonucleotide, and 2.5 U of TaqDNA Polymerase. The cycling conditions are 2 min at 94 °C for the initiation, followed by 39 cycles of 30 sec at 94 °C, 45 s at 60 °C, 45 s at 72 °C and a final extension of 10 min at 72 °C. Amplified DNA is analyzed on the TapeStation System, an automated electrophoresis platform, by using the D1000 ScreenTape for the analysis of DNA from 35 to 1000 bp. Two reference strains for Malassezia were used as a positive control for the multiplex PCR (purchased from Centraalbureau voor Schimmelcultures Fungal Biodiversity Centre, Utrecht, The Netherlands). Additional DNA from skin FFPE specimens derived from patients suffering from Pityriasis versicolor served as positive control (cordial Univ. Prof. B. Zelger from the Department of Dermatology ad Venereology of the Medical University of Innsbruck).

### Appendix A.1.4. Immunohistochemistry

Sections with a thickness of 1 µm were prepared on TOMO adhesion microscopic slides (Matsunami, Washington, DC, USA).

Immunohistochemistry was performed with commercially available kits, and pre-tests were performed to decide whether Standard Kits or the more sensitive Elite Kits had to be used for the different markers. For this establishment of the methods, specimens from normal or diseased non-malignant pancreatic tissue were used.

Briefly, FFPE tissue sections were stained by the ABC method using appropriate specific antibodies. Sections were deparaffinized and rehydrated, followed by the blockage of endogenous peroxidase activity and unspecific binding.

The antigens were detected without pretreatment or after unmasking by proteinase K or heat. Sections were incubated with the primary antibody, control sections with an unspecific antibody of the same isotype and were visualized with a biotinylated secondary antibody. The bound antibody was detected using different chromogens with signal amplification by the corresponding ABC kits. The sections were counterstained by hematoxylin, and the slides were mounted in Permount medium.

HE-staining was routinely performed following the standard house protocol at the Department of Pathology, Neuropathology and Molecular Pathology, University of Innsbruck.

A semiquantitative score (from 0 to 3 points) was defined to determine the level of expression of different complement proteins within the specimens, depending on the intensity of the color signal at the histopathology. For each patient, analyses were conducted both on specimens of the main lesion as well as on perilesional pancreatic tissue. Furthermore, the expression of each complement protein was considered in separated compartments, i.e., in pancreatic epithelial cells, in the intercellular stroma and in the inflammatory cell compartment. For each specimen, the mean intensity score for each protein was considered one time in regard to the center of the lesion and one time in relation to perilesional tissue.

### Appendix A.1.5. Statistical Analysis

The clinical data were entered into an electronic database (Excel 2011; Microsoft Corp., Redmond, WA, USA) and analyzed using SPSS software (version 24.0; SPSS Inc., Chicago, IL, USA). Categorical data were summarized using absolute and relative frequencies. Quantitative data were summarized using the mean. In the case of non-normally distributed data, non-parametric methods were used for evaluation (chi-square test for categorical, Mann–Whitney U test for continuous data). A *p*-value of <0.05 was considered statistically significant for all statistical analyses.

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
