# Peer review of "Complement and Fungal Dysbiosis as Prognostic Markers and Potential Targets in PDAC Treatment"

_curroncol, doi:10.3390/curroncol29120773_

Round 1

Reviewer 1 Report

The authors have done a comprehensive work in illustrating the roles of Complement system and fungal dysbiosis in modulating the TME and their association in PDAC tumor biology.

The study is so overwhelming where it addresses two distinct topics that serve as biomarker for PDAC. The later part of the manuscript looks more like a research article. This could be a two different articles.  

Reviewer 2 Report

I should state that I am neither a complement or a fungal expert and can therefore only review this article as a general pancreatologist.

This review article on complement and fungal dysbiosis in PDAC is timely and relevant. It is, however, a slightly unusual review article because it also presents new data and this element is a substantial part of the paper. If this is not in conflict with the policy of the journal, I have nothing against it, but I must emphasize that the new data, as indeed admitted by the authors (l. 478), are preliminary and based on a VERY limited data set.

The authors are clearly complement experts and, as far as I can read, the complement biochemistry described and discussed seems credible and is competently explained.

[1] I am, however, concerned that the critical TME is not well introduced and that the data on both the complement involvement and the fungal dysbiosis are not related or discussed in relation to the various cell types that are such a critical component of the TME. The key general figure (Fig. 2), although signposted as dealing with “Complement in tumor microenvironment”, just shows the pancreas as a whole without any attempt to illustrate the TME. The almost consistent ambiguity, throughout the review article, concerning WHERE various processes occur is a problematic feature of this article. Some references are related to the endocrine rather than the exocrine pancreas! Clearly, there will in many cases be insufficient information about the locality of particular processes, and this will have to be admitted, but never even to discuss this critical point does not seem right.

[2] When the TME is first introduced (l. 69 – 72) there should be a brief account of its constituents. Specifically, it is a serious omission that the pancreatic stellate cells are never mentioned anywhere in this review article. They are increasingly seen as a critical component in pancreatic pathophysiology, and particularly in relation to pancreatic inflammation (see: Physiol Rev 101, 1691-1744, 2021, which is highly relevant in the context of this review article. When, for example, at l. 114, an action of thrombin is discussed, it would be relevant to mention that there is evidence that thrombin acts specifically on activated pancreatic stellate cells (see again: Physiol Rev 101, 1691-1744, 2021).

Round 2

Reviewer 2 Report

The authors have done a good revision job, taking care of all the critical points I had made. The paper has been markedly improved and is, in my opinion, now ready for acceptance.